# Paired Pulse Suppression and Prepulse Inhibition in Patients with Autism Spectrum Disorder and Attention Deficit Hyperactivity Disorder

**DOI:** 10.3390/brainsci15101052

**Published:** 2025-09-27

**Authors:** Dai Suzuki, Eishi Motomura, Kazuki Hisatomi, Yusuke Nakayama, Takayasu Watanabe, Motohiro Okada, Koji Inui

**Affiliations:** 1Department of Neuropsychiatry, Mie University Graduate School of Medicine, Tsu 514-8507, Japan; dsky@clin.medic.mie-u.ac.jp (D.S.); motomura@clin.medic.mie-u.ac.jp (E.M.); okadamot@clin.medic.mie-u.ac.jp (M.O.); 2Department of Clinical Laboratory, Mie University Hospital, Tsu 514-8507, Japan; clt-kaz-hisatomi@med.mie-u.ac.jp (K.H.); yusuke-n@med.mie-u.ac.jp (Y.N.); takayasu-w@med.mie-u.ac.jp (T.W.); 3Department of Functioning and Disability, Institute for Developmental Research, Aichi Developmental Disability Center, Kasugai 480-0392, Japan; 4Section of Brain Function Information, National Institute for Physiological Sciences, Okazaki 444-8553, Japan

**Keywords:** auditory evoked potentials, change-related potentials, sensory gating, sensory inhibition

## Abstract

**Objectives:** We examined whether sensory inhibition was altered in patients with autism spectrum disorder (ASD) and attention deficit hyperactivity disorder (ADHD) as deficits in the neural inhibition are considered to be involved in both disorders. **Methods:** By using auditory change-related brain potentials as the test response, paired pulse suppression (PPS) and prepulse inhibition (PPI) were compared among healthy controls (*n* = 57), patients with ASD (*n* = 22), and ADHD (*n* = 8). The test change-related response was elicited by an abrupt sound pressure increase in a continuous sound. In the PPS experiment, two 15 dB change stimuli were given 600 ms apart to elicit two change-related responses. In the PPI experiment, the test stimulus of a 10 dB increase and prepulse of a 2 dB increase were given with an interval of 50 ms. Evoked potentials were recorded from Cz referenced to the linked mastoids. **Results:** PPS differed significantly among the three groups, with a significantly lower value for ADHD than controls. PPI was significantly lower for ASD and ADHD than normal controls. **Conclusions:** Since these two measures are thought to represent changes in circuit excitability due to preceding stimuli via GABAergic transmission, the present results support the idea that dysfunction of the GABAergic system contributes to the etiology of these disorders. The present results showed that the pattern of dysfunction differed between the two disorders and suggest that measurements of inhibitory function may be able to differentiate between the two disorders.

## 1. Introduction

The output of a neural circuit is controlled by the balance of activity between excitatory pyramidal neurons and inhibitory interneurons. Because dysfunction of inhibitory interneurons is thought to be involved in the pathogenesis of several diseases, including developmental disorders, it is necessary to know how the inhibitory function works in each patient with such diseases. For this, measurements of synaptic events elicited by interneurons through GABA, the major inhibitory transmitter in the brain, are required. However, direct observation of inhibitory interneuron activity requires invasive methods such as patch clamping, which is not applicable to humans. As an indirect method for measuring inhibitory function in humans, some studies use paired pulse suppression (PPS) or prepulse inhibition (PPI) that are based on the fact that a sensory stimulus inhibits the response to a following stimulus [1,2]. It is called PPS when the two successive stimuli are identical, and called PPI when the first stimulus is too weak to evoke a reliable response. Usually, different sensory stimuli and different test responses are used for PPS and PPI: two successive click sounds are used to evoke auditory evoked potentials for PPS, while a short burst of loud white noise evokes blink reflexes for PPI. In both paradigms, PPS and PPI values indicate how strongly the first stimulus affects the response to the second stimulus.

In our group, auditory change-related potentials have been used to observe PPS and PPI [3,4,5]. The change-related potential is an event-related potential specifically elicited by abrupt changes in a continuous sensory stimulus. Two identical change stimuli 600 ms apart elicit change-related responses twice, with the second response being smaller, i.e., PPS [4,6]. When a weak change stimulus (prepulse) precedes the test stimulus by 10–800 ms, the test response magnitude is smaller than when the test stimulus is presented alone, i.e., PPI [7]. PPS shows a peak of inhibition at conditioning-testing intervals (CTIs) of 600–700 ms [4], while PPI shows multiple peaks [7], suggesting that multiple temporally distinct mechanisms contribute to PPI, e.g., inhibition at 10–30 ms PTIs reflects early inhibitory postsynaptic potentials (IPSPs) produced by GABA_A_ receptor activation [8,9]. Thus, several inhibitory functions with distinct mechanisms can be tested using PPS and PPI.

A certain group of autism spectrum disorder (ASD) is believed to have increased excitation relative to inhibition [10,11,12], which matches the fact that ASD is frequently accompanied by epilepsy [13,14]. The rate of the comorbidity is approximately 30% [15]. To test the excitatory–inhibitory imbalance hypothesis of ASD, many studies have attempted to find changes in glutamate, the primary excitatory transmitter, and GABA in ASD. Studies using postmortem tissues have shown a reduction in GABA_A_ receptor and glutamic acid decarboxylase (GAD) density, which is consistent with decreased inhibition. Studies using magnetic resonance spectroscopy, positron emission tomography, and single photon emission tomography found in vivo that the level of GABA receptors or GABA concentrations was reduced in ASD (for reviews see [16,17,18]). It is of note that these findings in ASD patients are not restricted to a brain area, suggesting regional non-specific alterations at least in part in this disease. Some recent studies have also shown an important role of excitatory/inhibitory imbalance in attention deficit/hyperactivity disorder (ADHD). Findings supporting this include altered levels of GABA concentration or GABA receptor density in the brain (for reviews see [19,20]). The excitation/inhibition imbalance, if present, may be specific to a subgroup of ASD or ADHD and contribute to heterogeneity of these diseases, or the imbalance may be ubiquitous and contribute to a variety of clinical manifestations in sensory, memory, language, and social systems [10]. In the present study, we examined PPS and PPI in patients with ASD and ADHD and compared them with normal controls to know whether disruption in inhibitory function in the sensory system is present in patients with these disorders.

## 2. Methods

This study was carried out in accordance with the Declaration of Helsinki on 30 patients with developmental disorders, aged 6–38 (22.3 ± 9.4) years, and 57 healthy controls (HCs), aged 6–38 (19.1 ± 9.3) years. The study was approved in advance by the Ethics Committee of Aichi Developmental Disability Center (approved number, 16–03). Written informed consent was obtained from all participants. None of the HCs had any history of mental or neurological disorders or substance abuse in the last two years. They were free of medication at testing. Outpatients were enrolled from the Department of Psychiatry, Mie University Hospital. All were diagnosed with ASD/ADHD and met all criteria outlined in the DSM-V (Diagnostic and Statistical Manual of Mental Disorders, 5th edition). Participants with a history of convulsions other than febrile convulsions were excluded. Participants had normal hearing as assessed by an audiometer (AA-71, Rion, Tokyo, Japan). When the hearing test could not be conducted, we confirmed that participants identified the change in sound pressure from the baseline. There were two experiments, Experiment 1 (Exp1) for PPS and Experiment 2 (Exp2) for PPI, and they were conducted in that order.

### 2.1. Sound Stimuli

For both experiments, repeats of a 25 ms pure tone (800 Hz, rise and fall times of 5 ms) were used as described elsewhere [21]: 80 repeats of the 25 ms tone at 65 dB SPL in Exp1 and 24 repeats at 70 dB in Exp2 (Figure 1). For the change stimulus eliciting change-related potentials, two consecutive 25 ms tones at 80 dB were used. In Exp1, there was only one stimulus that had two identical sound pressure changes at 1100 and 1700 ms after the onset of the stimulus, which elicited the change-related potentials twice, 600 ms apart. In Exp2, there were four stimuli: the control stimulus without a change, a test alone stimulus with the change stimulus of 80 dB at 350 ms, a prepulse alone stimulus with a weak change stimulus of 72 dB at 300 ms, and a prepulse + test stimulus with the prepulse at 300 ms and test at 350 ms. Therefore, the prepulse–test interval was 50 ms or the inter-stimulus interval was 25 ms. Sound stimuli were created using a PC (Windows XP, 32 bit) and delivered through a headphone.

### 2.2. Recordings

Auditory evoked potentials were recorded using an EMG/EP measuring system (MEB-2300 or EEG-1214, Nihon Kohden, Tokyo, Japan). The sampling rate was 1000 Hz. An exploring electrode was placed at Cz referenced to the linked P9-P10 [3]. A pair of electrodes were placed on the supra- and infra-orbit of the right eye to reject trials with eye blinking. Impedance for all the electrodes was less than 5 kΩ. The analysis window was 1000 to 2000 ms from the onset of the auditory stimulus in Exp1, and from 100 ms to 600 ms after the stimulus onset in Exp2. The EMG/EP measuring system recorded only the specified epochs and automatically averaged them. The analog filter was 0.5–100 Hz. Trials with activity larger than 100 μV at any electrode were automatically excluded from averaging.

### 2.3. Procedures

The experiments were conducted in a quiet, electrically shielded room. Participants sat on a chair and watched a silent movie on a screen 1.5 m in front of them during the recording. They were instructed to concentrate on the movie and ignore the stimuli. In Exp1, the stimulus was presented at a trial–trial interval of 2200 ms. After the averaging reached 120 epochs, Exp2 was immediately initiated. In Exp2, the four stimuli were presented randomly at a trial–trial interval of 800 ms. For each stimulus, at least 120 epochs were averaged.

### 2.4. Analysis

An offline filter of 0.9–35 Hz was applied to recorded signals for analyses [5]. The abrupt change in a continuous tone elicited triphasic response at Cz with a positivity at approximately 50 ms (P50), negativity at 100 ms (N100), and positivity at 200 ms (P200) in adult participants [3]. For the amplitude of the response, a P50-N100-P200 peak-to-peak-to-peak amplitude was used [20], which was calculated by the following formula: (P50-N100 peak-to-peak amplitude + N100-P200 peak-to-peak amplitude)/2. However, in the participants 15 years old and younger, the triphasic response was absent, or weak if present, and the response composed of a positivity at 150 ms (P150) and negativity at 250 ms (N250). Therefore, P150-P250 peak-to-peak amplitude was used in such cases.

In Exp1, the response to the second change stimulus was reduced relative to the first response. The degree of PPS was calculated by the following formula: (first response amplitude—second response amplitude)/first response amplitude × 100. The amplitude of the first response and PPS were used for statistical analyses. In Exp2, subtraction procedures were necessary to obtain the test alone and prepulse + test response [3]. The test alone response was obtained by subtraction of the response to the standard stimulus from the response to the test alone stimulus. The prepulse + test response was obtained by subtracting the prepulse alone response from the response to the prepulse + test stimulus. The amplitude was measured using the difference waveforms. The amplitude of the test response was reduced by the presence of the prepulse. The degree of PPI was calculated by the following formula: (test alone amplitude − prepulse + test amplitude)/test alone amplitude × 100. In addition to the test alone amplitude and PPI, the onset response was analyzed. The onset response was obtained by averaging all trials. Similarly to the change-related response, the peak-to-peak-to-peak or peak-to-peak amplitude was measured.

Because of the different shapes of the test response waveform for adults and children, these data were treated as separate groups. Thus, statistical analyses were performed using Group (HC, ASD, and ADHD) × Waveform (ADULT and YOUNG) 2-way ANOVA. Pairwise comparisons were made using Bonferroni’s correction when necessary. Normal distribution of the data of each group was confirmed by the Kolmogorov–Smirnov test. The significance of differences was set at *p* < 0.05. All statistical analyses were performed using SPSS ver. 24.

## 3. Results

In both experiments, all adult participants 18 years old and older showed a triphasic test response, while participants 15 years old and younger showed a biphasic test response. Out of 87 participants, 56 showed the triphasic test response: 40 HCs (17 females and 23 males, 18–38 years) and 16 patients (8 females and 8 males, 18–38 years). The age did not differ significantly between the two groups (27.7 ± 4.9 vs. 27.0 ± 6.4 years, *p* = 0.65). On the other hand, 31 participants showed the biphasic test response: 17 HCs (6 females and 11 males, 8–15 years) and 14 patients (6 females and 8 males, 6–15 years), without a significant age difference (9.8 ± 3.1 vs. 10.9 ± 2.9 years, *p* = 0.35).

### 3.1. Experiment 1

Two abrupt changes in sound pressure elicited the change-related response twice, with the latter response being smaller (Figure 2). The amplitude of the first response and PPS value are listed in Table 1. The first response amplitude did not differ significantly among the three groups (*p* = 0.37) but was significantly larger for YOUNG than ADULT (*p* = 1.6 × 10^−5^). PPS for the second response differed significantly among the three groups (F_2,81_ = 3.6, *p* = 0.033, partial η^2^ = 0.08), and between ADULT (31.3%) and YOUNG (20.0%) (F_1,81_ = 5.4, *p* = 0.023). Following paired comparisons showed that ADHD patients had significantly lower PPS (15.7%) than HCs (32.5%) (*p* = 0.011). The PPS value of the ASD group (28.7%) was intermediate between HC and ADHD.

### 3.2. Experiment 2

Like in Exp1, the amplitude of the test response was significantly smaller for ADULT than YOUNG (*p* = 1.4 × 10^−4^), and was significantly different among groups (F_2,81_ = 3.4, *p* = 0.039, partial η^2^ = 0.08). As compared to HCs (7.0 µV), patients with ASD (6.0 µV) and ADHD (5.3 µV) showed smaller test responses, but the difference between any pair was not significant. PPI differed significantly among the groups (F_2,81_ = 8.5, *p* = 4.6 × 10^−4^, partial η^2^ = 0.17) but not between YOUNG and ADULT (*p* = 0.89). Post hoc tests showed that patients with ASD (6.6%) had significantly weaker inhibition than HCs (27.9%, *p* = 0.002). PPI of patients with ADHD (8.1%) was also smaller than HCs, but the difference did not reach the significant level (*p* = 0.095). The grand-averaged waveforms in Figure 3 show deficits in PPI in patients with developmental disorders. The onset response was triphasic in ADULT while biphasic in YOUNG, similar to the test response (Figure 4). The amplitude of the onset response differed significantly among the groups (F_2,80_ = 3.2, *p* = 0.046, partial η^2^ = 0.074) and between ADULT and YOUNG (F_1,80_ = 88.7, *p* = 1.3 × 10^−14^, partial η^2^ = 0.53). In addition, there was a significant Group × ADULT/YOUNG interaction (F_2,80_ = 3.8, *p* = 0.026, partial η^2^ = 0.087). Following analyses showed that the onset response amplitude differed significantly among the three groups in ADULT (F_2,52_ = 25.1, *p* = 2.4 × 10^−8^) but not in YOUNG. In ADULT, the amplitude was the highest in HC (3.2 µV) followed by ASD (1.2 µV) and ADHD (0.7 µV). Paired comparisons showed a significant difference between HC and ASD (*p* = 1.1 × 10^−6^) and between HC and ADHD (*p* = 0.0004).

### 3.3. Treatment

Of the 30 patients, 18 were on medication: neuroleptics (*n* = 6), anti-depressants (*n* = 6), atomoxetine (*n* = 4), lemborexant (*n* = 4), ramelteon (*n* = 3), anti-epileptics (*n* = 2), and melatonin (*n* = 2). When patients with and without medication were compared, PPI (22.2 ± 17.4 and 27.7 ± 26.4%) and PPS (5.5 ± 37.5 and 5.8 ± 25.9%) did not differ significantly (*p* > 0.53). Similar analyses on each drug showed that any drug did not significantly affect PPI and PPS (*p* > 0.07, not corrected for multiple comparisons). Although the present patients did not have epilepsy, 10 patients showed abnormal or epileptiform EEG findings: spikes in the frontal (*n* = 2), temporal (*n* = 2), or central area (*n* = 2), 6 Hz phantom spike and wave (*n* = 3), and frontal irregular delta waves (*n* = 1). However, both PPI (29.2 ± 15.4%) and PPS (6.2 ± 22.2%) of the 10 patients did not differ significantly from PPI (21.9 ± 26.2%) and PPS (9.0 ± 27.9%) of the remaining 20 patients with normal EEG (*p* > 0.42).

## 4. Discussion

This study examined whether PPI and PPS were altered in patients with ASD and ADHD. Compared to healthy controls, PPI was significantly lower in both ASD and ADHD patients, and PPS was significantly lower in ADHD patients. Thus, the present results confirm the deficits in inhibition in these disorders, but also distinguish between the two. These findings are consistent with the idea that deficits in inhibition or excitation/inhibition imbalance play a role in the etiology of these disorders [10,18,19].

PPI of the change-related response peaks at several CTIs from 10 to 800 ms suggests that several mechanisms with different temporal profiles contribute to it [7,8]. PPI at the CTI of 25–50 ms in the present study is thought to involve the early part of GABA_A_-mediated IPSPs, because benzodiazepines affect the inhibition at this timing and the time course of the inhibition coincides with the early phase of IPSPs indued by GABA_A_ receptor activation [7,8,9]. Inhibitory control through GABA_A_ receptors is related to various functions including precise timing of pyramidal cell firing, sedation, memory, and anxiety [22,23,24,25]. Therefore, impairment of PPI in patients may contribute to the occurrence or symptoms of the disorder. The present finding is consistent with previous studies using in vivo neuroimaging and postmortem brain tissues showing altered GABAergic systems in ASD [16,18] and ADHD [19,20]. For example, recent studies using magnetic resonance spectroscopy (MRS) found lower GABA levels across brain areas in ASD patients compared with normal controls [26]. Results of postmortem studies showed reduced GABA receptor density or GAD in ASD patients [15]. Hong et al. reported reduction in the GABA_A_ receptor α2 subunit in supragranular layers of prefrontal cortex areas in ASD [27]. In ADHD children, GABA concentration is reduced in the motor cortex, striatum, and thalamus [19]. Thus, the present study functionally confirmed that patients with developmental disorders have GABAergic dysregulation and that it may contribute to the pathophysiology of these disorders.

PPS in the present study represents long-latency inhibition with the CTI of 600 ms. Although a weak prepulse could suppress a following test response at this CTI, the inhibition threshold is higher than that at shorter intervals; inhibition could not be observed in some subjects [7]. Therefore, in this study, paired pulse paradigm was used to obtain stable inhibition. Under the paradigm used in this study, the degree of inhibition was greatest at CTIs of 600–700 ms [4]; inhibition at shorter intervals exerts less inhibition. Therefore, at least a part of the inhibition is not due to depressing synapse that is stronger at shorter intervals [28]. As the responsible mechanism, slow IPSPs produced by Martinotti cells are one of the candidates [7]. Martinotti cells work to suppress excessive firing of pyramidal cells [29]. The results of the preset study showed that patients with ADHD had significantly weaker PPS than HCs, which may be correlated with ADHD symptoms such as impulsiveness. However, it was not clear whether this finding was specific to ADHD, as ASD patients showed a similar tendency, and there was no significant difference between ADHD and ASD. It may suggest differences in the degree of impairment. This finding is comparable in adults and children (Table 1) and contrasts with previous findings that intracortical inhibition measured using transcranial magnetic stimulation (TMS) applied to the primary motor cortex [30] is reduced in children with ADHD but not in adults with ADHD [31]. Because motor hyperactivity, a core symptom of ADHD declines with age [32], differences in intracortical inhibition between children and adults with ADHD appear to correlate with motor cortical dysregulation. On the other hand, the deficit in PPS in this study may reflect GABAergic dysfunction stable from childhood to adult.

The above-mentioned association between deficits in inhibition in the motor cortex and motor performance in ADHD is further strengthened by the finding that reduced intracortical inhibition in the motor cortex correlates with ADHD diagnosis and symptom severity and also reflects motor skill development in ADHD children [33]. In addition, the degree of intracortical inhibition was correlated with the GABA level in the motor cortex measured by MRS [34], supporting the relationship between symptoms and GABAergic function in related cortical areas. However, a question then arises as to whether dysfunction in cortical excitability in ADHD is specific to the motor cortex or present throughout the brain [35]. Likewise, it is unclear whether reduced PPI or PPS in the present study in the patient group is limited to the auditory cortex or common across brain regions. In this regard, previous postmortem and in vivo imaging studies showed reduction in GABA receptor density or reduction in GABA concentration in various brain regions including the sensory cortex, anterior cingulate gyrus, hippocampus, and cerebellum [16,17]. In various animal models of ASD, the number of parvalbumin-positive (PV) interneurons is reduced [36]. PV interneurons represent the largest class of GABAergic interneurons, accounting for about 40% of cortical interneurons [37], and densely connect to surrounding pyramidal cells [38]. In ADHD, GABA concentration is reduced in the somatosensory cortex, motor cortex [39], striatum [40], and anterior cingulate cortex. Thus, altered GABAergic function in ADHD is possibly ubiquitous throughout the brain. Furthermore, correlation between clinical symptoms and deficits in the GABA system in a cortical area other than the motor cortex has been reported. Smaller increases in GABA concentrations in the anterior cingulate cortex (ACC) measured under the Stroop task correlate with impaired attentional control in ADHD patients relative to control subjects [41]. In female ADHD patients, GABA level in ACC is reduced, and the GABA level was correlated to impulsiveness [42]. Thus, dysfunction of GABAergic transmission is common throughout the brain in these diseases, and each symptom may be related to the function of the affected brain region. In the present study, reduced inhibition in patients with developmental disorders may correlate with sensory dysfunction such as hypersensitivity in the auditory system. However, in our previous study, the degree of PPS was correlated in the somatosensory and auditory systems [6], suggesting that similar mechanisms are involved in PPS in both systems, supporting the view of a general impairment of the inhibitory system. Taken together, the PPI and PPS may represent a wide range of inhibitory functions in the brain that are not restricted to specific sensory areas and can therefore be used as a functional measure of a patient’s inhibitory system.

The present paradigm is non-invasive, with no concerns about effects on brain function [43]. Participants need only watch the video during the recording, which is relatively unburdening, and so, it would be useful for clinical patients. However, the paradigm is not appropriate for patients with auditory hypersensitivity. In such cases, a similar paradigm using the trigeminal blink reflex can be used [9,44,45,46]. The present study did not examine the relationship between the inhibitory ability and symptoms, which needs further studies with a larger sample size. Comparisons of inhibition among several different methods may be fruitful for clarifying the mechanism of PPS and PPI.

## 5. Conclusions

Deficits in PPI or PPS in patients with developmental disorders may represent ubiquitous alteration of the inhibitory control of the brain not restricted to the auditory cortex and, therefore, may be a marker of the mechanism underlying clinical symptoms including the language and social problems not restricted to the sensory system. However, if the present findings represent alterations in all interneurons of patients with developmental disorders, then deficits in inhibition can occur in inhibitory neurons, which would lead to increases in inhibition. Therefore, the interpretation is not straightforward. Nevertheless, it is clear that the patients in this study differed functionally from healthy controls. In addition, some items in the testing could distinguish ASD and ADHD, suggesting different contributions of GABAergic abnormalities to these diseases. Further studies with a larger sample and information on clinical characteristics are needed to clarify the differences in inhibitory function among ASD, ADHD, and ASD + ADHD [47,48,49,50].

## Figures and Tables

**Figure 1 brainsci-15-01052-f001:**
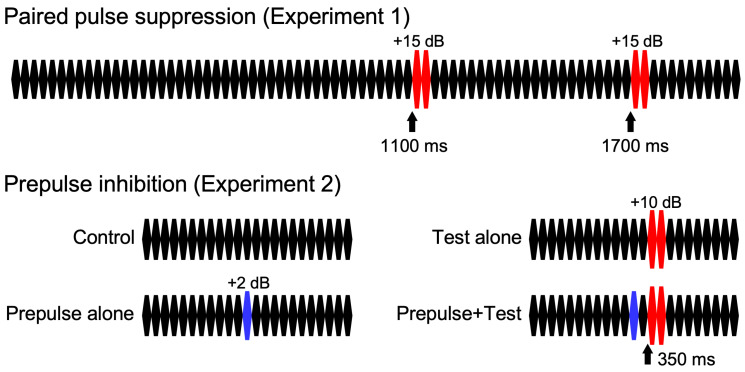
Sound stimuli. Sound stimuli consist of 25 ms pure tone repetition.

**Figure 2 brainsci-15-01052-f002:**
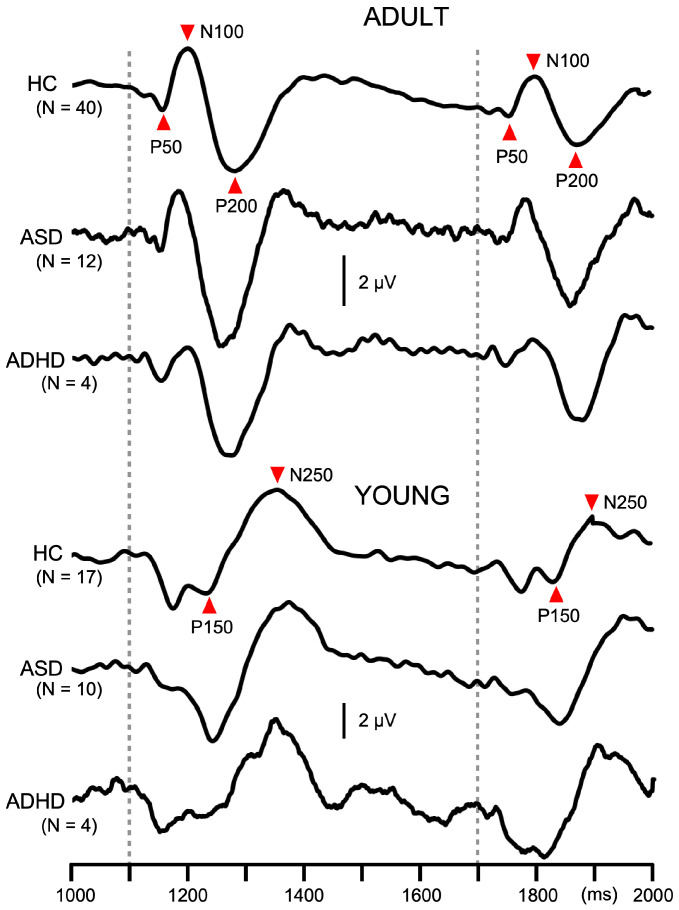
Average waveforms across participants for paired pulse suppression in Experiment 1. Traces show auditory change-related potentials at Cz in the paired pulse suppression experiment. In each participant, 120 epochs were averaged. Dotted lines indicate the timing of the abrupt change in sound pressure. ADULT, participants with the triphasic change-related response; YOUNG, participants with the biphasic change-related response; ASD, autism spectrum disorder; ADHD, attention deficit hyperactivity disorder.

**Figure 3 brainsci-15-01052-f003:**
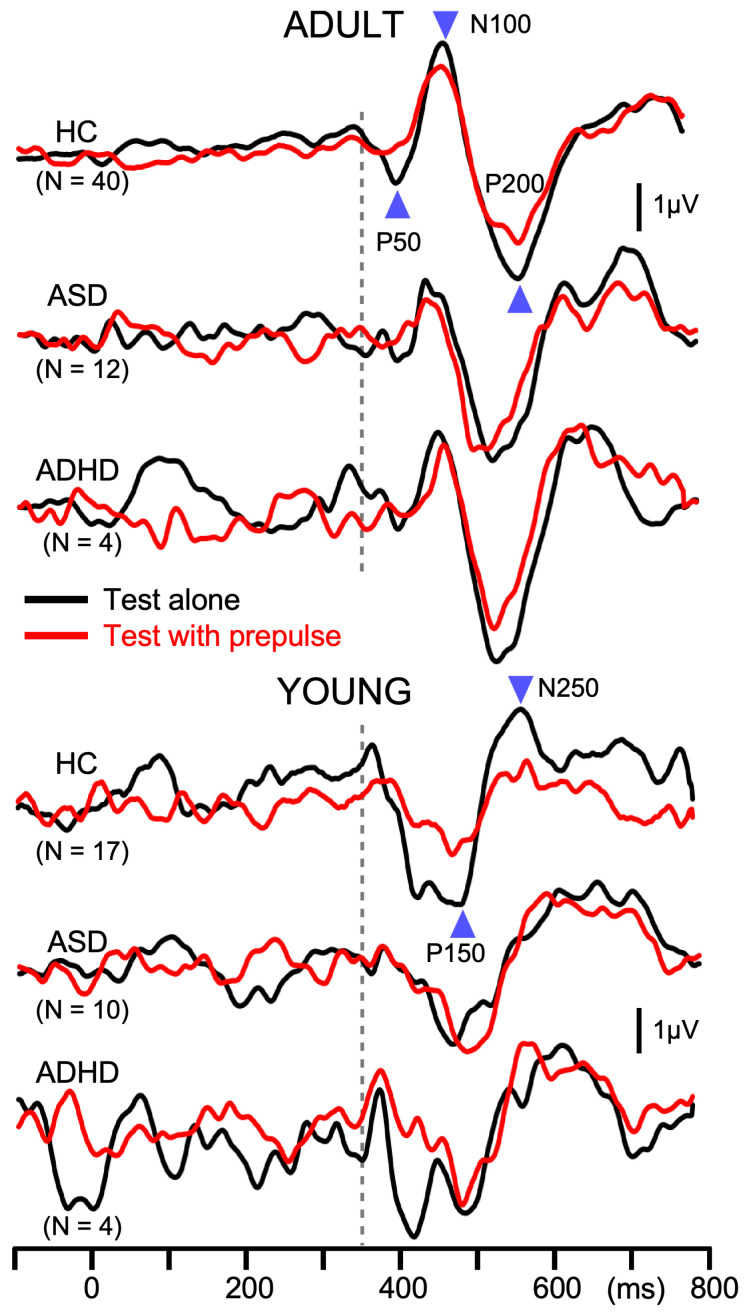
Average waveforms across participants for prepulse inhibition in Experiment 2. Waveforms show auditory change-related potentials at Cz in the prepulse inhibition experiment. In each participant, 120 epochs were averaged. The abrupt change in sound pressure occurred at 350 ms, indicated by the dotted line. Black and red lines indicate responses to the test alone stimulus and to test + prepulse stimulus, respectively.

**Figure 4 brainsci-15-01052-f004:**
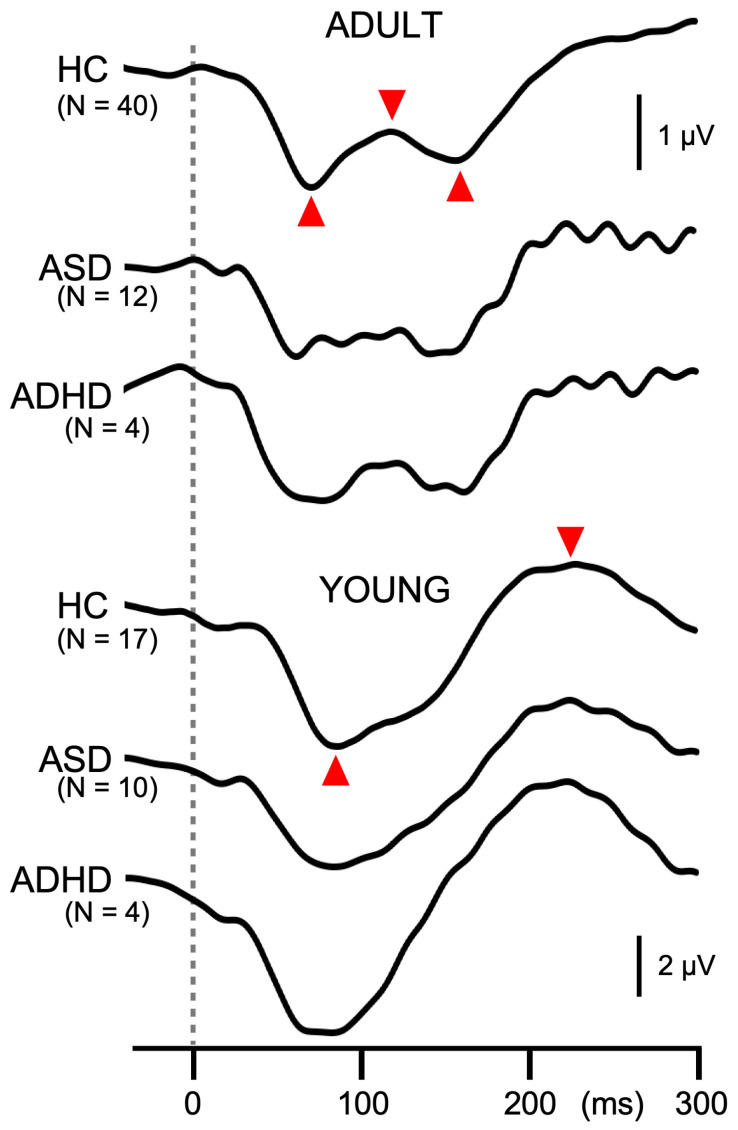
Grand-averaged waveforms of the onset response in Experiment 2. Waveforms show the onset response at Cz averaged across all conditions. The dotted line indicates the onset of stimuli. Similarly to the change-related response, peak-to-peak-to-peak or peak-to-peak amplitude (indicated by red triangles) is used to measure the onset response.

**Table 1 brainsci-15-01052-t001:** The amplitude of the test responses and their inhibition.

	Experiment 1			Experiment 2		
	Age (Years)	S1 (µV)	PPS (%)	Test (µV)	PPI (%)	Onset (µV)
ADULT						
HC	27.7 (4.9)	5.4 (2.1)	37.4 (12.8)	5.9 (2.3)	24.5 (24.2)	3.2 (1.2)
ASD	27.3 (6.6)	5.6 (2.7)	33.5 (14.8)	4.0 (2.2)	4.5 (23.6)	1.2 (0.6)
ADHD	26.0 (6.3)	3.3 (2.0)	22.9 (4.5)	3.9 (2.4)	15.8 (25.0)	0.7 (0.4)
YOUNG						
HC	9.8 (3.1)	8.9 (5.4)	27.7 (13.1)	9.6 (4.2)	36.0 (18.3)	7.3 (3.3)
ASD	10.8 (3.0)	10.6 (4.8)	23.9 (23.7)	8.3 (4.2)	10.4 (33.2)	6.5 (3.1)
ADHD	11.0 (2.9)	9.0 (4.8)	8.5 (44.3)	6.7 (4.0)	0.5 (42.0)	9.2 (4.6)

S1, response to the first stimulus. PPS, paired pulse suppression. PPI, prepulse inhibition. HC, healthy control. Data are shown in the mean (standard deviation).

## Data Availability

The data presented in this study are available on request from the corresponding author due to privacy reasons.

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
