# Peer review of "Paired Pulse Suppression and Prepulse Inhibition in Patients with Autism Spectrum Disorder and Attention Deficit Hyperactivity Disorder"

_brainsci, 2025, doi:10.3390/brainsci15101052_

Round 1
Reviewer 1 Report
Comments and Suggestions for Authors
This research explored PPS and PPI in patients with ASD and ADHD and compared them with healthy controls to determine whether a disruption in inhibitory function in the sensory system is present in patients with these disorders.
This is well-developed research; the manuscript was clearly described, and the statistical analysis was well documented.
The results will be of interest to the scientific community in this research field. I only have minor comments to improve the clarity of this manuscript.
Minor comments:
-The number of samples for averaging each EP and the total number of subjects for averaging could be included in the figure legends. Figure legends could also specify which type of EP was recorded.
-Units for age should be included in Table 1.
-In Table 1, the numbers within parentheses should be defined in the Table legend (are SD or SEM?)
Reviewer 2 Report
Comments and Suggestions for Authors
This work studies paired pulse suppression and prepulse inhibition in developmental disorders namely ASD and ADHD and compares the response against healthy controls. The study design is fine and it is good that authors treated young and adult participants as separate groups for analysis while ensuring the age didn't change significantly across the groups. As a reader, the method section seemed incomplete and didn't provide all the details necessary to understand the results 100%. I recommend that authors add the following details to the method and analysis section:
- How many channels were recorded using the acquisition device (EMG/EP system and where were they placed on the scalp/head?
- What kind of preprocessing and referencing was done while recording the signal and were any channels rejected from analysis?
- Does the acquisition device record raw EEG or only the evoked potentials?
Additional comments:
- What does grand average mean in figures 2 and 3, were the waveforms averaged across all or certain channels per participant and then across participants?
- In figure 3, what do the two line plots (red and black) represent? The manuscript doesn't describe what they are and neither does the figure caption.
Round 2
Reviewer 2 Report
Comments and Suggestions for Authors
No more comments. Authors have addressed the concerns in the revised manuscript.